# Absolute Positioning and Orientation of MLSS in a Subway Tunnel Based on Sparse Point-Assisted DR

**DOI:** 10.3390/s20030645

**Published:** 2020-01-23

**Authors:** Qian Wang, Chao Tang, Cuijun Dong, Qingzhou Mao, Fei Tang, Jianping Chen, Haiqian Hou, Yonggang Xiong

**Affiliations:** 1China University of Geosciences, NO.29 Xueyuan Road, Beijing 100083, China; 3001100114@cugb.edu.cn (Q.W.); 2015600009@cugb.edu.cn (J.C.); 2Geophysical Exploration Academy of China Metallurgical Geology Bureau, NO.139 Sunshine North Street, Baoding 071051, China; 3Zhengyuan Geophysical Co., Ltd., NO.139 Sunshine North Street, Baoding 071051, China; 4Beijing Urban Construction Exploration & Surveying Design Research Institute CO., LTD, Beijing 100101, China; houhaiqian@cki.com.cn; 5Urban Intelligent Perception & Precision Measurement Engineering Technology Center, Wuhan University, No.129 Luoyu Road, Wuhan 490079, China; 6State Key Laboratory of Information Engineering in Surveying, Mapping and Remote Sensing, Wuhan University, No.129 Luoyu Road, Wuhan 430079, China; qzhmao@whu.edu.cn; 7Wuhan Metro Bridge and Tunnel Management Co., Ltd., NO.22 Qinyuan Road, Wuhan 430000, China; tangfei@wuhanrt.com; 8Wuhan Hirail Profiling Technology Co., Ltd, Wuhan 430060, China; ygxiong@hirail.cn

**Keywords:** GNSS-denied, mobile laser scanner system, cloud points, dead reckoning, local coordinate system

## Abstract

When performing the inspection of subway tunnels, there is an immense amount of data to be collected and the time available for inspection is short; however, the requirement for inspection accuracy is high. In this study, a mobile laser scanning system (MLSS) was used for the inspection of subway tunnels, and the key technology of the positioning and orientation system (POS) was investigated. We utilized the inertial measurement unit (IMU) and the odometer as the core sensors of the POS. The initial attitude of the MLSS was obtained by using a static initial alignment method. Considering that there is no global navigation satellite system (GNSS) signal in a subway, the forward and backward dead reckoning (DR) algorithm was used to calculate the positions and attitudes of the MLSS from any starting point in two directions. While the MLSS passed by the control points distributed on both sides of the track, the local coordinates of the control points were transmitted to the center of the MLSS by using the ranging information of the laser scanner. Then, a four-parameter transformation method was used to correct the error of the POS and transform the 3-D state information of the MLSS from a navigation coordinate system (NCS) to a local coordinate system (LCS). This method can completely eliminate a MLSS’s dependence on GNSS signals, and the obtained positioning and attitude information can be used for point cloud data fusion to directly obtain the coordinates in the LCS. In a tunnel of the Beijing–Zhangjiakou high-speed railway, when the distance interval of the control points used for correction was 120 m, the accuracy of the 3-D coordinates of the point clouds was 8 mm, and the experiment also showed that it takes less than 4 h to complete all the inspection work for a 5–6 km long tunnel. Further, the results from the inspection work of Wuhan subway lines showed that when the distance intervals of the control points used for correction were 60 m, 120 m, 240 m, and 480 m, the accuracies of the 3-D coordinates of the point clouds in the local coordinate system were 4 mm, 6 mm, 7 mm, and 8 mm, respectively.

## 1. Introduction

The subway has gradually become the main tool for urban transportation because of its convenience and speed. With the continuous progress of urbanization, the operation mileage of subway systems has also increased. Periodic inspections of subway tracks, tunnels, and other related facilities are indispensable for ensuring operational safety. However, the structure of subway facilities is complex, the data of inspection are numerous, and the time available for inspection is short. At present, the number of inspection vehicles and the inspection technology cannot meet the needs of daily operations maintenance. Thus, new inspection vehicles and technologies need to be put into use as soon as possible [1,2,3].

A mobile laser scanning system (MLSS) utilizes a laser scanner as a measurement sensor and an inertial measurement unit (IMU) as the core sensor of the positioning and orientation system (POS) [4,5]. The 3-D point cloud of the scanned object can be quickly acquired by using a mobile measurement system (MMS). The utilization of MLSS can help maintenance personnel to rapidly reconstruct the 3-D shape of subway tunnels and related facilities [6,7,8,9]. Based on this, minor deformations and defects of subway-related facilities can be identified accurately, including tunnel diameter convergence measurement, boundary detection, ellipticity analysis, analysis of the slab staggering of duct pieces, leakage detection, seepage detection, lining detection, and so forth [3,10,11,12,13]. Furthermore, by comparing the absolute and relative position changes of multi-period observation results, we can identify the characteristics and corresponding changes of convergence, gauge, the slab staggering of duct pieces, leakage, and other diseases in the subway tunnel, based on this, the safety state of subway tunnel can be obtained. Absolute accuracy is mainly used to ensure the accuracy of disease location and the feasibility of multi-stage data comparison. Only when we get the absolute position of the point cloud of the tunnel, the subsequent diseases inspection can be carried out.

The accuracy and precision of point clouds obtained by using MLSS is affected by many factors, the most important of which is the accuracy of the POS [14]. A pure inertial navigation system (INS) has the disadvantage of error accumulation [15]. Generally, a combination of multiple navigation technologies is used to overcome the shortcomings of a single navigation technique. The best mode of an integrated navigation system is to select complementary navigation systems to assist each other, and it is better to have absolute and relative positioning techniques at the same time [16]. Absolute positioning techniques provide absolute position reference and correct the accumulated errors caused by relative positioning techniques. The accuracy of an absolute positioning technique determines the best positioning accuracy that can be obtained by an integrated navigation system, while a relative positioning technique provides high-frequency and rich state information regarding the carrier [17,18,19]. The classical techniques are the global navigation satellite system (GNSS) + INS, GNSS + dead reckoning (DR), GNSS + INS + DR, and so forth. However, there are no GNSS signals in a subway environment to provide external reference information to correct positioning errors and unify the coordinate systems [20,21,22]. Therefore, the main research problem relates to developing a method to obtain accurate positions and attitudes of MLSS by using IMU as the core sensor of POS in subway environments with no GNSS signals.

There are some studies on the inspection and maintenance of rail transit facilities using inertial integrated navigation technology, but most of them are under the conditions of GNSS signals [23,24,25,26,27,28,29]. These methods cannot use external information to correct inertial navigation errors in a subway environment with no GNSS signals.

Relative positioning techniques available in rail transit environments include INS, odometers, axle counters, and so forth. With the assistance of laser/vision data, the simultaneous localization and mapping (SLAM) method can be utilized to complete positioning and attitude determination without GNSS signals [30,31]. However, the accuracy cannot meet the requirements of the inspection of subway tunnels; thus, other absolute positioning information is still necessary for the high-precision inspection of subway tunnels. The absolute positioning information available in a subway environment includes map matching (MM), intermittent responders, coordinates of control points, and so forth [21]. However, only the accuracy of coordinates of control points can meet the requirements of the inspection of subway tunnels.

Many scholars and research institutions have carried out studies based on this. For example, absolute measurement technology uses the total station to set up stations freely at certain distance intervals and calculates the absolute coordinates of the total station by measuring multiple third-class control points (CPIII); then, high-precision coordinates are transferred to the MMS to correct the accumulated errors. The representative static track inspection trolley—Graduate Representative Programme (GRP) 1000 from the Switzerland Emberger Technology Company—is mainly used to measure the static geometry of a track. The absolute positioning accuracy of a track can reach up to 3 mm. In addition, the GEDO CE track inspection trolley from the Trimble Navigation Company is used for the high-precision control of double-block ballastless tracks. These two products are also the most widely used in track detection work at present, but their measurement efficiency is relatively low, and the detection distance of each skylight time is about 300 m. Further, operators are required to have special knowledge and skills.

Li Q. used stereo cameras to connect a track control network with MLSS [32], and Mao Q., Zhang L., and Li Q. et al. utilized a laser scanner to transfer the coordinates of railway track control points to MLSS [33]. They used the coordinates of railway track control points to correct the accumulated errors in order to improve efficiency and accuracy [34]. However, the geodetic coordinates of the initial point of MLSS have to be known, and the coordinates of point clouds in the LCS cannot be directly obtained by data fusion.

To address the existing problems in the inspection of subway tunnels, in this work, we studied and designed a set of navigation solutions and point cloud data fusion methods, and the 3-D state information of MLSS was calculated by the forward/reverse DR algorithm. By scanning railway control points with a laser scanner, the coordinates of the control points were transferred to the center of the MLSS. For data postprocessing, a four-parameter transformation method was utilized to convert the position and attitude of the MLSS from a navigation coordinate system (NCS) to an LCS, which corrected the error of POS information and reduced the time cost of point cloud data fusion.

The rest of the paper is organized as follows: Section 2 presents all the steps and related principles and formulas involved in this method. Section 3 reports the specific experimental information and results. Section 4 concludes this paper.

## 2. The Principles of the Method

### 2.1. The Whole Procedure of the Method 

This paper presents a method for how to accurately obtain the absolute coordinates of laser points by using a MLSS. The implementation process of this algorithm is shown in Figure 1.

For a MLSS using IMU as its core component for navigation, obtaining the positions and attitudes of the MLSS under a targeted coordinate system so that we can use the positions and attitudes to transform the coordinates of laser points directly from the laser scanner coordinate system (LSCS) to the LCS is the main challenge.

The first step is to carry out the initial alignment process to obtain the 3-D state information of the starting time, including the position, velocity, and attitude. Based on this, the forward and backward DR algorithm can be utilized to calculate the 3-D positions, velocities, and attitudes of a MLSS under designated time intervals during the whole inspection process. Presently, the position and attitude of a MLSS are still referenced to an NCS (usually, the World Geodetic System 1984, WGS-84), and they need to be transformed into a LCS. The error correction and localization of the MLSS’s positions and attitudes is merged into one process by using the sectional four-parameter transformation method, and the coordinates of the control points are needed to calculate the corresponding parameters. The last step is to use the MLSS’s corrected and localized positions and attitudes to transform the coordinates of point clouds into a LCS, and such coordinates have both absolute and relative position information, meaning that the follow-up specific inspection work can be conducted. The objective of this study was to demonstrate a method that can acquire the 3-D state information of an MLSS in a GNSS-denied environment, so that we can obtain the high precision absolute coordinates of massive point clouds, which is only part of the point cloud application; thus, the content of the point cloud data fusion method is not discussed.

### 2.2. The Initial Alignment Method

The initial alignment is performed to determine the attitude of the vehicle relative to the NCS [35]; only by completing this step can the DR process be performed. The initial alignment only requires knowing the latitude of the starting point, not the longitude [36]. So, the coordinates of control points are used to roughly calculate a latitude value, and the longitude value is set at will.

The static initial alignment method is adequate for the acquisition of the initial attitude of a MLSS in the inspection of subway tunnels. Static initial alignment is mainly divided into two steps: coarse and fine alignments. Coarse alignment is used to obtain the rough value of the initial attitude of the body coordinate system relative to the NCS. Fine alignment is used to further estimate the misalignment angle through the error propagation theory of INS to further improve the accuracy of the initial attitude [37].

Current theoretical approaches of coarse alignment include vector attitude determination, analytical coarse alignment, and indirect coarse alignment. Coarse alignment algorithms are adequate for use, while research on fine alignment algorithms is still ongoing [38]. 

In this study, we utilized the Kalman filter to estimate the misalignment angle. The position remains unchanged and the velocity is zero in the static alignment process [39]. Based on this, the initial alignment equation is obtained by simplifying the error propagation equation of INS, as shown in formulas (1) and (2):(1)$$\dot{\phi }=\phi \times \omega_{ie}^{n}-\varepsilon^{n},$$
(2)$$\delta {\dot{v}}^{n}=f_{sf}^{n}\times \phi +\nabla^{n},$$
$$\varepsilon^{n}=\left[\begin{array}{c} \varepsilon_{E}\\ \varepsilon_{N}\\ \varepsilon_{U} \end{array}\right]=\left[\begin{array}{c} C_{11}\varepsilon_{x}^{b}+C_{12}\varepsilon_{y}^{b}+C_{13}\varepsilon_{z}^{b}\\ C_{21}\varepsilon_{x}^{b}+C_{22}\varepsilon_{y}^{b}+C_{23}\varepsilon_{z}^{b}\\ C_{31}\varepsilon_{x}^{b}+C_{32}\varepsilon_{y}^{b}+C_{33}\varepsilon_{z}^{b} \end{array}\right],\;\nabla^{n}=\left[\begin{array}{c} \nabla_{E}\\ \nabla_{N}\\ \nabla_{U} \end{array}\right]=\left[\begin{array}{c} C_{11}\nabla_{x}^{b}+C_{12}\nabla_{y}^{b}+C_{13}\nabla_{z}^{b}\\ C_{21}\nabla_{x}^{b}+C_{22}\nabla_{y}^{b}+C_{23}\nabla_{z}^{b}\\ C_{31}\nabla_{x}^{b}+C_{32}\nabla_{y}^{b}+C_{33}\nabla_{z}^{b} \end{array}\right],$$
where ϕ is the misalignment angle, $\dot{\phi }$ is the change rate of the misalignment angle, $f_{sf}^{n}$ is the specific force, and $\delta {\dot{v}}^{n}$ is the change rate of the velocity error; they are all referenced to NCS. $\varepsilon^{n}$ is the drift of the gyroscope, and $\nabla^{n}$ is the drift of the accelerometer; they are also referenced to NCS and can be deemed as random constant values if $\varepsilon^{b}={\left[\begin{array}{ccc} \varepsilon_{x}^{b} & \varepsilon_{y}^{b} & \varepsilon_{z}^{b} \end{array}\right]}^{T}$ and $\nabla^{b}={\left[\begin{array}{ccc} \nabla_{x}^{b} & \nabla_{y}^{b} & \nabla_{z}^{b} \end{array}\right]}^{T}$ are also constant values. $C_{b}^{n}$ is the matrix that transforms the coordinate from the body frame (it is the coordinate system of the IMU in this paper) to the NCS; it is also a constant value in static initial alignment mode [40].

The approximation is $f_{sf}^{n}\approx -g^{n}={\left[\begin{array}{ccc} 0 & 0 & g \end{array}\right]}^{T}$ in static mode (g is the value of gravity). Based on this, the formula of fine alignment is
(3)$$\left\{ \begin{array}{l} {\dot{\phi }}_{E}=\omega_{U}\phi_{N}-\omega_{N}\phi_{U}-\varepsilon_{E}\\ {\dot{\phi }}_{N}=-\omega_{U}\phi_{E}-\varepsilon_{N}\\ {\dot{\phi }}_{U}=\omega_{N}\phi_{E}-\varepsilon_{U}\\ \delta {\dot{v}}_{E}=-g\phi_{N}+\nabla_{E}\\ \delta {\dot{v}}_{N}=g\phi_{E}+\nabla_{N}\\ \delta {\dot{v}}_{U}=\nabla_{U} \end{array}\right.,$$
where ω is the angular rate. The last equation ($\delta {\dot{v}}_{U}=\nabla_{U}$) is unrelated to the other equations, which means that the error in the upward direction can be neglected because it does not decrease the accuracy of the estimated misalignment angle. The state space model is shown in formula (4):(4)$$\left\{ \begin{array}{l} \dot{X}=FX\\ Z=HX \end{array}\right.$$

The random constant drift of the gyroscope and accelerometer is added to the state variables:$$X={\left[\begin{array}{cccccccccc} \phi_{E} & \phi_{N} & \phi_{U} & \delta v_{E} & \delta v_{N} & \varepsilon_{E} & \varepsilon_{N} & \varepsilon_{U} & \nabla_{E} & \nabla_{N} \end{array}\right]}^{T}.$$

The coefficient matrix of the state variables is as follows:$$F=\left[\begin{array}{cccccccccc} 0 & \omega_{U} & -\omega_{N} & 0 & 0 & -1 & 0 & 0 & 0 & 0\\ -\omega_{U} & 0 & 0 & 0 & 0 & 0 & -1 & 0 & 0 & 0\\ \omega_{N} & 0 & 0 & 0 & 0 & 0 & 0 & -1 & 0 & 0\\ 0 & -g & 0 & 0 & 0 & 0 & 0 & 0 & 1 & 0\\ g & 0 & 0 & 0 & 0 & 0 & 0 & 0 & 0 & 1\\ & & & & 0_{5\times 10} & & & & & \end{array}\right].$$

The coefficient matrix of the measurement equation is as follows:$$H=\left[\begin{array}{cccccccccc} 0 & 0 & 0 & 1 & 0 & 0 & 0 & 0 & 0 & 0\\ 0 & 0 & 0 & 0 & 1 & 0 & 0 & 0 & 0 & 0 \end{array}\right].$$

The outputs of velocity can be set as the new information. Then, the recursive solution is carried out until the result converges, and the misalignment angle can be obtained accurately.

### 2.3. The INS/Odometer Integrated Navigation Algorithm

In Figure 2, ∆s is the distance increment measured by the odometer; $\theta^{\prime \prime }$, $P^{\prime \prime }$, and $V^{\prime \prime }$ are the attitude, position, and velocity, respectively, calculated by pure INS; and $P^{\prime }$ and $V^{\prime }$ are the position and velocity, respectively, calculated by DR. The differences between positions calculated by INS and DR are used as the new information of the Kalman filter [41,42,43,44].

The propagation equation of position error δp calculated by INS is:(5)$$\delta \dot{p}=M_{v}\delta v+M_{p}\delta p,$$
$$M_{v}=\left[\begin{array}{ccc} 0 & 1/R_{M} & 0\\ 1/R_{N}\cos L & 0 & 0\\ 0 & 0 & 1 \end{array}\right],\;M_{pp}=\left[\begin{array}{ccc} 0 & 0 & -v_{N}/R_{M}^{2}\\ v_{E}\tan L/R_{N}\cos L & 0 & -v_{E}/R_{N}^{2}\cos L\\ 0 & 0 & 0 \end{array}\right],$$
where $R_{M}$ and $R_{N}$ are the curvature radii of the prime vertical and meridian, respectively, and L is the latitude.

The propagation equation of position error calculated by DR is
(6)$$\delta {\dot{p}}_{D}\;=\;M_{aD}\phi_{D}+M_{pD}\delta p_{D}+M_{TD}T_{D},$$
$$M_{aD}=M_{vD}(v_{D}^{n}\times ),M_{TD}=M_{vD}M_{TD},$$
where $v_{D}^{n}\times$ is the skew-symmetric matrix of $v_{D}^{n}$; $M_{vD}$ has the same form as $M_{v}$ and $M_{pD}$ has the same form as $M_{pp}$, except that the elements in the matrix are calculated by DR.
$$M_{vTD}=v_{D}\left[\begin{array}{ccc} -C_{13} & C_{11} & C_{12}\\ -C_{23} & C_{21} & C_{22}\\ -C_{33} & C_{31} & C_{32} \end{array}\right],\;T_{D}=\left[\begin{array}{c} \alpha_{\theta }\\ \alpha_{\psi }\\ \delta K_{D} \end{array}\right],$$
where $\alpha_{\theta }$ and $\alpha_{\psi }$ are the installation error of pitch and heading angles of the odometer coordinate system relative to the INS coordinate system, respectively, and $\delta K_{D}$ is the scale factor error of the odometer’s outputs. $\alpha_{\theta }$, $\alpha_{\psi }$, and $\delta K_{D}$ are usually added to state variables in the GNSS/INS-integrated navigation process in order for them to be solved together; however, there are no GNSS signals in a subway environment. In [32] and [33], the coordinates of control points in a subway environment are used as external reference information to estimate the parameters $\alpha_{\theta }$, $\alpha_{\psi }$, and $\delta K_{D}$. However, the number of control points is small and the distance interval is 60 m, which provides too little information to estimate that many unknown parameters with good accuracy. Further, the solution results change considerably at the place at which there are control points [45,46,47,48].

There are other parameters such as δl (lever arm: vector of the center of the odometer coordinate system relative to the center of the INS coordinate system) and δt (time synchronization error of the odometer relative to the INS). Since the structure of the instrument is very strong, $\alpha_{\theta }$, $\alpha_{\psi }$, $\delta K_{D}$, and δl are quite stable; so, $\alpha_{\theta }$, $\alpha_{\psi }$, $\delta K_{D}$, and δl should be calibrated before detection work, instead of adding them to state variables and estimating them during the filtering process, as this is more beneficial to the results. In this study, the calibration of $\alpha_{\theta }$, $\alpha_{\psi }$, and $\delta K_{D}$ was carried out on a straight track with a good GNSS signal by comparing the positions and velocities calculated by the INS/odometer and GNSS. The exact values of $\alpha_{\theta }$, $\alpha_{\psi }$, and $\delta K_{D}$ could be obtained, and δl was obtained through the design parameters of the instrument.

δt is difficult to calibrate because it needs a much more accurate time reference system. Thus, it was added to the state variables in this study:(7)$$\left\{ \begin{array}{l} \dot{X}=FX+GW^{b}\\ Z=HX+V \end{array}\right..$$

$V_{p}$ is the measurement noise. The vector of the state variables is
$$X={\left[\begin{array}{ccccccc} \phi^{T} & {\left(\delta v^{n}\right)}^{T} & {\left(\delta P\right)}^{T} & {\left(\nabla^{b}\right)}^{T} & {\left(\varepsilon^{b}\right)}^{T} & {\left(\delta P_{D}\right)}^{T} & \delta t \end{array}\right]}_{19\times 1}^{T}.$$
$$F=\left[\begin{array}{c} \begin{array}{l} \begin{array}{cccccc} M_{aa} & M_{av} & M_{ap} & -C_{b}^{n} & 0_{3\times 3} & \begin{array}{cc} 0_{3\times 3} & 0_{1\times 1} \end{array}\\ M_{va} & M_{vv} & M_{vp} & 0_{3\times 3} & C_{b}^{n} & \begin{array}{cc} 0_{3\times 3} & 0_{1\times 1} \end{array}\\ 0_{3\times 3} & M_{pv} & M_{pp} & 0_{3\times 3} & 0_{3\times 3} & \begin{array}{cc} 0_{3\times 3} & 0_{1\times 1} \end{array} \end{array}\\ \begin{array}{cccccc} M_{aD} & 0_{3\times 3} & \;0_{3\times 3} & \;0_{3\times 3} & 0_{3\times 3} & \begin{array}{cc} M_{pD} & 0_{1\times 1} \end{array} \end{array} \end{array}\\ 0_{19\times 1} \end{array}\right],\;G=\left[\begin{array}{c} \begin{array}{cc} -C_{b}^{n} & 0_{3\times 3}\\ 0_{3\times 3} & C_{b}^{n} \end{array}\\ 0_{13\times 6} \end{array}\right],$$
$$W^{b}={\left[\begin{array}{cccccc} w_{gx}^{b} & w_{gy}^{b} & w_{gz}^{b} & w_{ax}^{b} & w_{ay}^{b} & w_{az}^{b} \end{array}\right]}^{T}.$$

$p_{INS}$ and $p_{DR}$ are the theoretical values of the positions calculated by pure INS and DR, respectively:(8)$$P_{INS}+M_{v}C_{b}^{n}\delta l+M_{v}v^{n}\delta t=\;P_{DR},$$
where ${\bar{p}}_{INS}$ and ${\bar{p}}_{DR}$ are the measured values of positions calculated by pure INS and DR, respectively, $\delta p_{INS}$ and $\delta p_{DR}$ are the corresponding errors:(9)$$\{\begin{array}{c} {\bar{p}}_{INS}\;=\;p_{INS}+\delta p_{INS}\\ {\bar{p}}_{DR}=p_{DR}+\delta p_{DR} \end{array}.$$

Considering formula 8 and formula 9, and taking the difference between the positions calculated by INS and DR as the measurements, the observation equation is obtained as shown in formula (10):$${\bar{p}}_{INS}-{\bar{p}}_{DR}=-M_{v}C_{b}^{n}\delta l-M_{v}v^{n}\delta t+\delta p_{INS}-\delta p_{DR}.$$
(10)$${\bar{p}}_{INS}+M_{v}C_{b}^{n}\delta l-{\bar{p}}_{DR}=\delta p_{INS}-\delta p_{DR}-\;M_{v}v^{n}\delta t.$$

The coefficient matrix of the observation equation is:$$H=\left[\begin{array}{ccccc} 0_{3\times 6} & I_{3\times 3} & 0_{3\times 6} & -I_{3\times 3} & -M_{v}v^{n} \end{array}\right].$$

This is the model of the INS/odometer integrated navigation system. With this model, the correct value can be obtained by using the Kalman filter to estimate state errors and make feedback corrections.

### 2.4. Forward and Backward Dead Reckoning 

#### 2.4.1. Unification of the Odometer’s Outputs

As described in Section 2.3, $\alpha_{\theta }$, $\alpha_{\psi }$, $\delta K_{D}$, and δl are calibrated in advance, so the transformation matrix between the body frame (INS frame) and the odometer coordinate system is:(11)$$C_{o}^{b}=I-(\alpha \times )=\left[\begin{array}{ccc} 1 & \alpha_{\psi } & -\alpha_{\gamma }\\ -\alpha_{\psi } & 1 & \alpha_{\theta }\\ \alpha_{\gamma } & -\alpha_{\theta } & 1 \end{array}\right]$$

The relationship between $v_{D}^{o}$ (the theoretical velocity in the odometer coordinate system) and ${\hat{v}}_{D}^{n}$ (the vector of the velocity measured by the odometer in the NCS) is:(12)$${\hat{v}}_{D}^{n}=C_{b}^{n}C_{o}^{b}(1+\delta K_{D})v_{D}^{o},$$
$$v_{D}^{o}=[\begin{array}{ccc} 0 & v_{D} & 0 \end{array}],$$
where $C_{o}^{b}$ is the transformation matrix that converts the coordinate from the odometer coordinate system to the body frame, and $C_{b}^{n}$ is the transformation matrix that converts the coordinate from the body frame to the NCS [49,50].

The relationship between the vector of the distance increment $\Delta S_{D}^{o}$ (the vector of the theoretical distance increment in the odometer coordinate system) and $\Delta {\hat{S}}_{D}^{n}$ (the vector of the distance increment measured by the odometer in NCS) is:(13)$$\Delta {\hat{S}}_{D}^{n}=C_{b}^{n}C_{o}^{b}(1+\delta K_{D})\Delta S_{D}^{o},$$
$$\Delta S_{D}^{o}=T\cdot v_{D}^{0}=T\cdot {\left[\begin{array}{ccc} 0 & V_{D} & 0 \end{array}\right]}^{T},\;\Delta {\hat{S}}_{D}^{n}=T\cdot {\hat{v}}_{D}^{n}={\left[\begin{array}{ccc} \Delta S_{N}^{n} & \Delta S_{E}^{n} & \Delta S_{U}^{n} \end{array}\right]}^{T},$$
where T is the sampling time interval.

#### 2.4.2. Forward DR

Formulas (14)–(16) show the position, velocity, and attitude update equation of forward DR:(14)$$\left\{ \begin{array}{l} L_{k}=L_{k-1}+\frac{\Delta S_{Nk-1}^{n}}{R_{M}+h_{k-1}}\\ \lambda_{k}=\lambda_{k-1}+\frac{\Delta S_{Ek-1}^{n}\sec L_{k-1}}{R_{N}+h_{k-1}}\\ h_{k}=h_{k-1}+\Delta S_{Uk-1}^{n} \end{array}\right.,$$
(15)$$v_{k}^{n}=v_{k-1}^{n}+T_{s}[C_{bk-1}^{n}f_{sfk}^{b}-(2\omega_{iek-1}^{n}-\omega_{enk-1}^{n})\times v_{k-1}^{n}+g^{n}],$$
(16)$$C_{bk}^{n}=C_{bk-1}^{n}(I+T_{s}\Omega_{nbk}^{b}),$$
where subscripts K−1 and K denote the order of calculation; $L_{k}$, $\lambda_{k}$, and $h_{k}$ are the latitude, longitude, and geodetic height, respectively; $\omega_{ie}^{n}$ is the earth rotation vector; $\omega_{en}^{n}$ is the relative attitude change rate of the NCS relative to the earth center coordinate system ($\omega_{ie}^{n}$ and $\omega_{en}^{n}$ are both in the NCS); $f^{b}$ is the specific force measured by the accelerometer in the body frame; $g^{n}$ is the gravity vector in the NCS; and $\Omega_{nbk}^{b}$ is the attitude change rate of the body frame relative to the NCS and is measured in the body frame [51].

#### 2.4.3. Backward DR

Formulas (17)–(19) show the position, velocity, and attitude update equation of backward DR:(17)$$\left\{ \begin{array}{l} L_{k-1}=L_{k}-\frac{\Delta S_{Nk-1}^{n}}{R_{M}+h_{k-1}}\approx L_{k}-\frac{\Delta S_{Nk-1}^{n}}{R_{M}+h_{k}}\\ \lambda_{k-1}=\lambda_{k}-\frac{\Delta S_{Ek-1}^{n}\sec L_{k-1}}{R_{N}+h_{k-1}}\approx \lambda_{k}-\frac{\Delta S_{Ek-1}^{n}\sec L_{k}}{R_{N}+h_{k-1}}\\ h_{k-1}=h_{k}-T_{s}v_{Uk-1}^{n}\approx h_{k}-\Delta S_{Uk-1}^{n} \end{array}\right..$$
(18)$$\left\{ \begin{array}{ll} v_{k-1}^{n} & =v_{k}^{n}-T_{s}[C_{bk-1}^{n}f_{sfk}^{b}-(2\omega_{iek-1}^{n}+\omega_{enk-1}^{n})\times v_{k-1}^{n}+g^{n}]\\ & \approx v_{k}^{n}-T_{s}[C_{bk}^{n}f_{sfk-1}^{b}-(2\omega_{iek-1}^{n}+\omega_{enk-1}^{n})\times v_{k-1}^{n}+g^{n}] \end{array}\right..$$
(19)$$C_{bk-1}^{n}=C_{bk}^{n}{\left(I+T_{s}\Omega_{nbk}^{b}\right)}^{-1}\approx C_{bk}^{n}(I-T_{s}\Omega_{nbk}^{b}).$$

The meaning of each parameter was discussed in Section 2.4.2. With the backward DR algorithm, the requirements for the known coordinates of the initial point are no longer necessary. Theoretically, any point in the trajectory can be used as the starting point for forward and backward DR to obtain the 3-D state information of the MLSS [51].

### 2.5. Conversion Method for Obtaining Absolute Coordinates 

#### 2.5.1. Traditional Coordinate Transformation Method

Usually, point clouds use geodetic coordinates because the results of INS/odometer-integrated navigation are referenced to the Earth center coordinate system. Thus, the coordinates of point clouds need to be transformed from a geodetic coordinate system (NCS) to an LCS (usually, the geodetic coordinate system is the Gauss projection coordinate system, and sometimes the system is rotated at a specific parameter) [14]. Figure 3 shows the coordinate transformation process before the improvement.

Formula (20) shows the principle of the transformation process from an LSCS to a geodetic coordinate system and then to an LCS:(20)$$X^{L}=T^{L}+m\cdot C_{n}^{L}(T^{n}+C_{b}^{n}(R_{l}^{b}\cdot X^{l}+T_{l}^{b})),$$
where $X^{L}$ is the coordinate in the LCS, $X^{l}$ is the coordinate in the LSCS, $T_{l}^{b}$ is the translation value from the LCS to the body frame, $R_{l}^{b}$ is the rotation matrix from the LSCS to the body frame, $T^{n}$ is the translation value from the body frame to the NCS, $C_{b}^{n}$ is the rotation matrix from the body frame to the NCS, $T^{L}$ is the translation value from the NCS to the LCS, m is the scale parameter, and $C_{n}^{L}$ is the rotation matrix from the NCS to the LCS.

#### 2.5.2. Improved Coordinate Transformation Method

The traditional coordinate transformation method needs three steps of point cloud transformation to obtain the coordinates of the point cloud in LCS; every step of point cloud transformation requires many calculations because of the huge amount of point cloud data, which greatly prolongs the data processing time. Based on the principles described in Section 2.5.1, we changed the MLSS’s positions and attitudes obtained through the INS/odometer-integrated navigation system by eliminating the step of transformation from the geodetic coordinate system to the LCS. Then, the three steps of transformation can be simplified into two steps when compared with the traditional method described in Section 2.5.1. Figure 4 shows the coordinate transformation process after the improvement.
(21)$$X^{L}=(T^{L}+m\cdot C_{n}^{L}T^{n})+m\cdot C_{n}^{L}C_{b}^{n}(R_{l}^{b}\cdot X^{l}+T_{l}^{b}),$$
where $\left(T^{L}+m\cdot C_{n}^{L}T^{n}\right)$ can be considered as the translation value $T_{b}{}^{L}$ from the body frame to the LCS, $T^{n}$ is actually the coordinates of the MLSS in the geodetic coordinate system, and $C_{n}^{L}C_{b}^{n}$ can be considered as the rotation matrix $C_{b}^{L}$ from the body frame to the LCS. The key is to obtain the values of $T^{L}$ and $C_{n}^{L}$.

#### 2.5.3. Transformation from NCS to LCS

Since the change is mainly in the horizontal direction, a four-parameter transformation method was used to calculate the transformation parameters between the NCS and the LCS.

The formula of the four-parameter transformation method is as follows:(22)$$\left[\begin{array}{c} x_{i}^{\prime }\\ y_{i}^{\prime } \end{array}\right]=\left[\begin{array}{c} T_{x}\\ T_{y} \end{array}\right]+\left[\begin{array}{cc} m\cos\Delta \psi & m\sin\Delta \psi \\ -m\sin\Delta \psi & m\cos\Delta \psi \end{array}\right]\left[\begin{array}{c} x_{i}\\ y_{i} \end{array}\right],$$
where the *x*-axis points to the east direction and the *y*-axis points to the north direction, $T_{x}$ and $T_{y}$ are the translation parameters, m is the scale factor, Δψ represents the rotation parameters, $x_{i}$ and $y_{i}$ are the original coordinates, and $x_{i}^{\prime }$ and $y_{i}^{\prime }$ are the coordinates of the targeted coordinate system [52,53]. 

Using the four-parameter transformation method is like bringing the trajectory from DR and the trajectory constrained by control points close to each other.

The solution process of these four parameters is as follows:

For convenience of calculation, $a=T_{x}$, $b=T_{y}$, c=mcosΔψ, and d=msinΔψ. Then, the coordinate transformation formula suitable for solving the four parameters is as follows:(23)$$\begin{array}{l} x_{i}^{\prime }=a+x_{i}c+y_{i}d\\ y_{i}^{\prime }=b+y_{i}c-x_{i}d \end{array}.$$

By using the adjustment method based on the four-parameter transformation method with at least two control points, the error equation can be obtained as follows:(24)$$\left[\begin{array}{c} v_{x1}\\ v_{y1}\\ v_{x2}\\ v_{y2}\\ \vdots \\ v_{xn}\\ v_{yn} \end{array}\right]=\left[\begin{array}{cccc} 1 & & x_{1} & y_{1}\\ & 1 & y_{1} & -x_{1}\\ 1 & & x_{2} & y_{1}\\ & 1 & y_{1} & -x_{2}\\ \vdots & \vdots & \vdots & \vdots \\ 1 & & x_{n} & y_{n}\\ & 1 & y_{1} & -x_{n} \end{array}\right]\left[\begin{array}{c} a\\ b\\ c\\ d \end{array}\right]-\left[\begin{array}{c} x_{1}^{\prime }\\ y_{1}^{\prime }\\ x_{2}^{\prime }\\ y_{2}^{\prime }\\ \vdots \\ x_{n}^{\prime }\\ y_{n}^{\prime } \end{array}\right].$$

After the four parameters are calculated, the rotation matrix from NCS to LCS can be obtained as follows:$$C_{n}^{L}=\left[\begin{array}{ccc} m\cos\Delta \psi & m\sin\Delta \psi & 0\\ -m\sin\Delta \psi & m\cos\Delta \psi & 0\\ 0 & 0 & m \end{array}\right].$$

The rotation matrix $C_{b}^{L}$, which denotes the transformation from the body frame to the LCS, is as follows:$$C_{b}^{L}=C_{n}^{L}C_{b}^{n}=\left[\begin{array}{ccc} T_{11}^{\prime } & T_{12}^{\prime } & T_{13}^{\prime }\\ T_{21}^{\prime } & T_{22}^{\prime } & T_{23}^{\prime }\\ T_{31}^{\prime } & T_{32}^{\prime } & T_{33}^{\prime } \end{array}\right],$$
$$\begin{array}{ll} T_{11}^{\prime }= & m\cos\Delta \psi (\cos\gamma \cos\psi +\sin\gamma \sin\psi \sin\theta )\\ & +m\sin\Delta \psi (\sin\gamma \cos\psi \sin\theta -\cos\gamma \sin\psi )\\ T_{21}^{\prime }= & -m\sin\Delta \psi (\cos\gamma \cos\psi +\sin\gamma \sin\psi \sin\theta )\\ & +m\cos\Delta \psi (\sin\gamma \cos\psi \sin\theta -\cos\gamma \sin\psi )\\ T_{31}^{\prime }= & -m\sin\gamma \cos\theta \\ T_{12}^{\prime }= & m\cos\Delta \psi \sin\psi \cos\theta +m\sin\Delta \psi \cos\psi \cos\theta \\ T_{22}^{\prime }= & m\cos\theta (\cos\Delta \psi \cos\psi -\sin\Delta \psi \sin\psi )\\ T_{32}^{\prime }= & m\sin\theta \\ T_{13}^{\prime }= & m\cos\Delta \psi \sin\psi \cos\theta +m\sin\Delta \psi \cos\psi \cos\theta \\ T_{23}^{\prime }= & -m\sin\Delta \psi (\sin\gamma \cos\psi -\cos\gamma \sin\psi \sin\theta )\\ & -m\cos\Delta \psi (\sin\gamma \sin\psi +\cos\gamma \cos\psi \sin\theta )\\ T_{33}^{\prime }= & m\cos\gamma \cos\theta \end{array}$$
where θ, γ, and ψ are the pitch, roll, and yaw angles of the body frame relative to the NCS, respectively. The attitudes of the body frame relative to the LCS can be obtained by using the components of $C_{b}^{L}$, which are as follows:(25)$$\{\begin{array}{c} \theta^{\prime }=\arcsin({T^{\prime }}_{32})=\theta \\ \gamma^{\prime }=\arctan(-\frac{{T^{\prime }}_{31}}{{T^{\prime }}_{33}})=\gamma \\ \psi^{\prime }=\arctan(\frac{{T^{\prime }}_{21}}{{T^{\prime }}_{22}})=\Delta \psi +\psi \end{array},$$
where $\theta^{\prime }$, $\gamma^{\prime }$, and $\psi^{\prime }$ are the pitch, roll, and yaw angles of the body frame relative to the LCS, respectively. The pitch and roll angles are the same as in the previous steps, and the yaw angle is the sum of the value of the previous yaw angle and the rotation angle Δψ.

By using the improved positions and attitudes of the MLSS, the coordinates of point clouds can be directly converted from the body frame to the LCS.

## 3. Experiments and Discussions

This section shows the experimental results of the inspection work performed in a tunnel of the Beijing–Zhangjiakou high-speed railway and the Wuhan subway line. The former is intended to show the detailed operation process and principle involved, and the latter is intended to verify the reliability and accuracy of the instrument and the proposed algorithm in the actual working environment.

### 3.1. Information about MLSS

Information about the MLSS used in the experiments is shown in Figure 5.

The inspection trolley utilizes a three-wheel structure to ensure that it is always close to the rail surface during the working process. In this experiment, GNSS equipment was only used for timing. The performance of the laser scanner of the MLSS is shown in Table 1.

The performance of IMU of MLSS is shown in Table 2.

### 3.2. Experiment in a Tunnel of the Beijing–Zhangjiakou High-Speed Railway

#### 3.2.1. Experimental Conditions

The internal environment of the high-speed railway tunnel is the same as that of a subway because there is also no GNSS signal. Control points are distributed on both sides of the track at certain distance intervals; i.e., 60 m. The total length of the tunnel studied was about 5–6 km and the speed for the inspection was about 3–5 km/h, meaning about 1.5 h was required to complete the data acquisition process. As the MLSS moves forward, the laser scanner on MLSS keeps scanning; the reconstructed 3D point cloud includes the subway tunnel and its related facilities, such as control points. This experiment was carried out according to the method and procedure of the inspection of subway tunnels.

#### 3.2.2. Experimental Results

The images generated by the data processing are discussed below.

Figure 6 shows the trajectory calculated by using the INS and odometer, but this trajectory was not corrected by control points. The green and red lines in Figure 6 represent the trajectory of the trolley calculated by the INS/odometer; the green part of the line means a relatively higher accuracy of 3-D coordinates when compared with the red part. The symbol of a red crucifix represents the control points, which are distributed on both sides along the track. The green flag represents the starting point and the end point. It can be seen that the starting and end points of the MLSS did not coincide with the control point, but when the inspection trolley passed through the control point, the coordinate of the control point could be transferred to the center of the body frame. Figure 8 will present the principle of how to extract the control points and transfer their coordinates to the center of the body frame. Then, the coordinates of the starting and end points could be obtained by using the forward and backward DR algorithm.

Figure 7 shows the strategy of using control points to correct the errors of position and orientation.

It can be seen from Figure 7 that the control points appear in pairs and are equally spaced (60 m). In the experiment of this paper, the control points used for correction were selected at equal distance intervals, and the distance interval was determined according to the situation. In Figure 7, the distance interval of control points selected for correction is 120 m.

Figure 8 shows how to extract control points from point clouds.

The approximate distance between the control points and the starting point of MLSS was recorded when we worked in the field. With the help of distance information and the point cloud processing software, we could fuse point clouds in the area of control points. After that, we got a result such as that shown in Figure 8. Figure 8 is enlarged to see the point cloud of the control point, and it will only take a few seconds to select one control point by clicking on it with a mouse. The coordinates of the control points in different coordinate systems (LSCS and LCS) can be calculated by using the position and attitude information without being corrected. The number of control points can be determined freely, and the distance interval was 120 m in this experiment.

After the control point was extracted, its coordinates in the LSCS reflected the coordinate difference between the control point and the center of MLSS, and its coordinates in the point clouds (the coordinate was obtained through calculation and relative to LCS) were compared with its real coordinates in the LCS to estimate the four parameters (mentioned in Section 2.5) to correct the positions and attitudes of the MLSS. Then, the point clouds do not need to be transformed from the body frame to a geodetic coordinate system/NCS and then transformed to LCS; instead, the positions and attitudes of the MLSS could be used again to directly transform all the other point clouds from the body frame to the LCS.

The point clouds in the LCS are shown in Figure 9.

Since the control points were distributed along a distance of 60 m and we only used half of them (extracting a pair of control points every 120 m) to calculate the four parameters, the other half were left for validation. When the whole solution process was completed, the coordinates of the laser point clouds were all relative to the LCS. The local coordinates of the control points (without having been used before) were compared to the coordinates of their corresponding laser points obtained through point cloud data fusion, and the absolute values of deviations in the horizontal coordinates, elevations, and 3-D positions are shown in Figure 10.

The abscissa represents the number of verification points, and the ordinate axis reflects the deviation in three directions. The specific statistical results of the deviations of these verification points are shown in Table 3. The red line represents the absolute values of deviations in the horizontal direction, the green line represents the absolute values of deviations in elevation, and the blue line represents the absolute values of deviations of the 3-D coordinates.

When there was no GNSS signal, the error correction and coordinate system transformation were carried out by using the control points at the distance interval of 120 m. The maximum deviations and the root mean square error (RMS) of the 3-D coordinates of the laser point cloud in the LCS were 14 mm and 8 mm, respectively, which meet the requirements for inspection results.

### 3.3. Experiments in the Inspection of Subway Tunnels 

The inspection work was carried out on Wuhan subway lines 2 and 6 for about one year. The detection work followed the method and procedure presented in this paper. Here, we introduced the experimental conditions and statistical results of the inspection.

#### 3.3.1. Experimental Conditions

The working conditions of the subway detection are shown in Figure 11.

All of the inspection work on the subway line was carried out at night, so the working environment was dark, but this did not affect the performance of the laser scanner.

Figure 12 shows an image of the point cloud of the subway tunnel, which displays its approximate geometry.

As showed in Figure 13, a paper target was placed next to the control point in order to make it convenient to determine the control point during postprocessing.

In order to save time, only a small number of point clouds around the control points were fused at first, and the laser points of the corresponding control points could be easily determined with the assistance of the paper target.

#### 3.3.2. Experimental Results and Discussions

The results were obtained by a statistical analysis of the data collected from the inspection work of Wuhan subway lines 2 and line 6 for about one year. Each inspection covered about 1–2 km, which is the distance between two subway stations.

Table 4 shows the position accuracy of the point clouds obtained by using the positions and attitudes of the MLSS, which had been corrected only by the coordinates of the control points near the start and end points (we need two control points that are not paired to correct the estimation and provide absolute reference information), and there was no other correction. The absolute accuracy of the 3-D coordinates was 2.3 cm, which shows the accuracy of the point clouds that can be obtained by using INS/odometer-integrated navigation with no control point corrections within a distance of 1–2km. Obviously, the accuracy of the point clouds was not high enough. However, the distance interval at which the point clouds should be corrected by the coordinates of the control points still needs to be verified.

Table 5 shows the position accuracy of the point clouds obtained by using the MLSS’s positions and attitudes that had been corrected by the coordinates of the control points in different distance intervals. From the comparison of Table 4 and Table 5, it can be seen that the accuracy was greatly improved in both the horizontal and elevation directions after the correction of the control points.

The accuracy of the 3-D coordinates of the point clouds obtained can reach up to 4 mm when the distance interval of the control point for correction is 60 m, and it can still reach up to 8 mm even when the distance interval for correction is 480 m. Thus, only one control point every 480 m is sufficient to achieve absolute detection accuracy at the millimeter level, which will greatly reduce the work intensity.

The approximate time taken in each step of the inspection work is as follows:(1)The data for a 5–6 km long tunnel can be collected in 1 h.(2)It takes about 5–10 minutes to fuse the point clouds of the areas around control points, and it only takes a few seconds to select a control point in the point cloud. In this way, we do not need to transform all the point clouds from a body frame to geodetic coordinate system/NCS, which has been described in 2.5.1; this greatly reduces the time required.(3)After that, the positions and attitudes of the MLSS will be corrected instantly with the algorithm given in Section 2.5.3.(4)Then, the coordinates of point clouds in the LCS can be directly obtained by using the positions and attitudes, which have been corrected; this step will take about 1 h.(5)Finally, the inspection of the diseases of a 5–6 km long tunnel based on point clouds will take about 1 h.

Thus, 4 h is enough for the whole inspection process of a 5–6 km long tunnel; in such a short time, the traditional inspection method cannot complete an inspection for such a long-distance, and some tunnel diseases cannot even be detected at all.

However, with the increase of the number of control points used for correction, the improvement of the plane position accuracy is not as obvious as the elevation. It is speculated that the error of the INS in the elevation direction is divergent, which is determined by the internal principle of the INS itself. The algorithm optimization of the elevation direction can be further expanded, so that it can still maintain high accuracy when there are few control points. Besides, the filtering and feature recognition algorithm of point clouds can be further studied to assist in the extraction of tunnel diseases in the process of data post-processing, which can reduce the workload.

## 4. Conclusions

The accuracy of the 3-D coordinates of the point clouds was 8 mm in a tunnel of the Beijing–Zhangjiakou high-speed railway when the distance interval of the control points used for correction was 120 m. The results of the inspection of Wuhan subway showed that when the distance intervals of the control points used for correction were 60 m, 120 m, 240 m, and 480 m, the accuracies of the 3-D coordinates of the point clouds were 4 mm, 6 mm, 7 mm, and 8 mm, respectively, in the LCS. The accuracy of the point cloud coordinates obtained by this method can completely meet the needs of the inspection of the subway. The results of the experiment in a tunnel of the Beijing–Zhangjiakou high-speed railway were basically the same as those in the Wuhan subway, and they both verified the accuracy and reliability of the proposed method.

Assisted by a small number of control points, this method can eliminate a MLSS’s dependence on GNSS signals, and for inspection work that requires coordinates to be obtained under the LCS, the positioning and orientation information can be used for point cloud data fusion to directly obtain the coordinates in the LCS, which greatly reduces the data processing time and also improves the accuracy of point clouds. Additionally, it takes less than 4 h to complete all the inspection work for a 5–6 km long tunnel, which is much more efficient than the traditional method. The experimental results proved the feasibility and high efficiency of the method proposed in this paper.

## Figures and Tables

**Figure 1 sensors-20-00645-f001:**
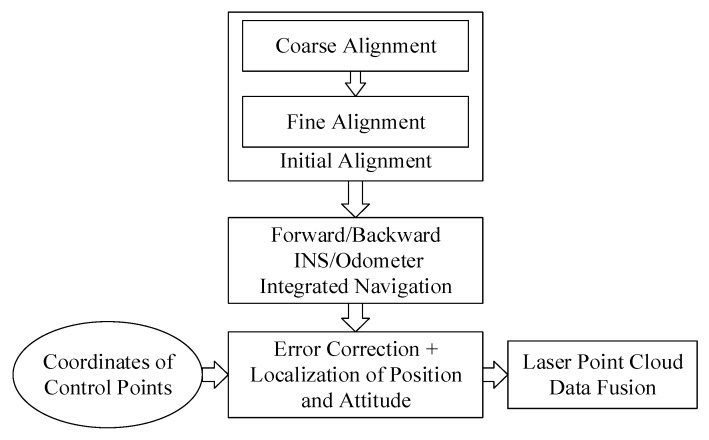
The overall flowchart of the method. INS: inertial navigation system.

**Figure 2 sensors-20-00645-f002:**
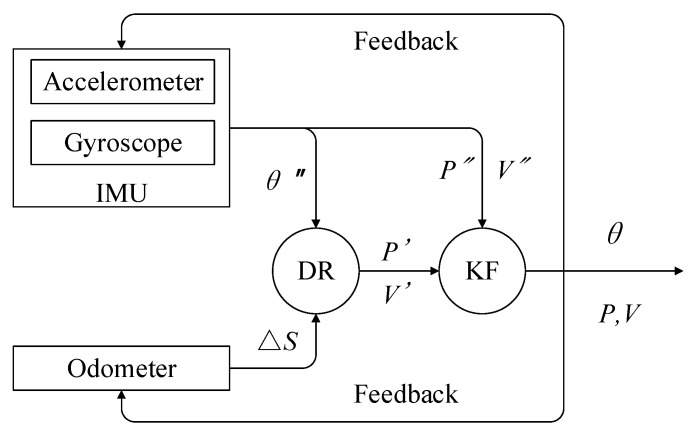
Structure charts of INS/odometer-integrated navigation. IMU: inertial measurement unit; DR: dead reckoning; KF: Kalman filter.

**Figure 3 sensors-20-00645-f003:**
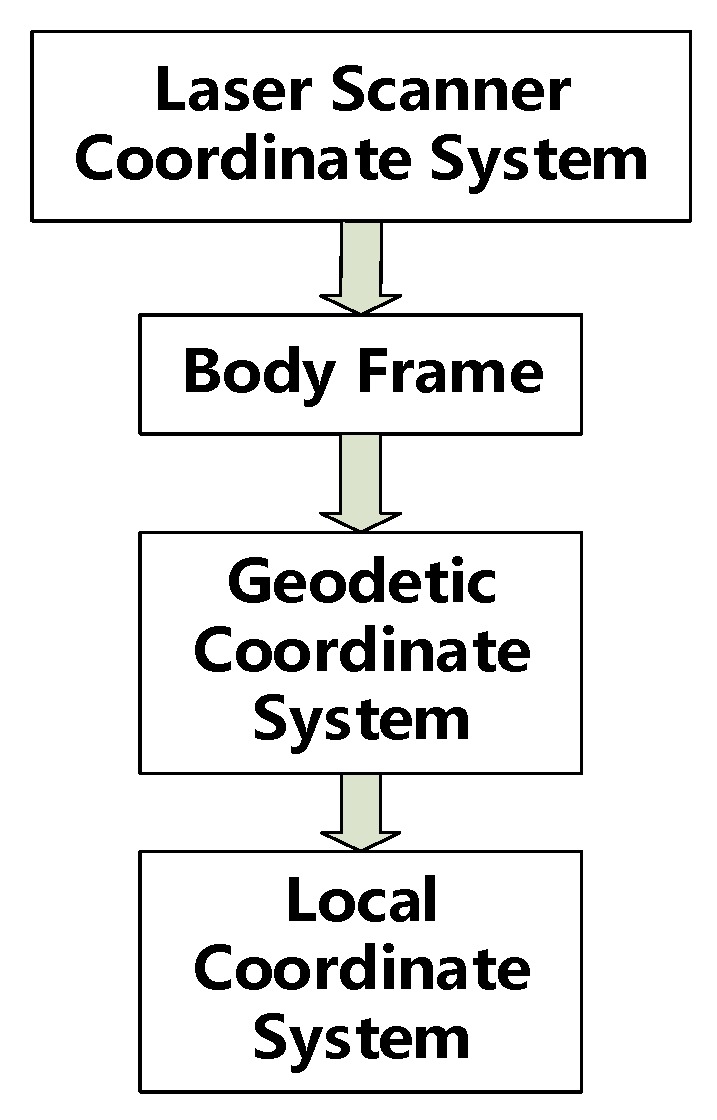
Procedure of previous coordinate transformation.

**Figure 4 sensors-20-00645-f004:**
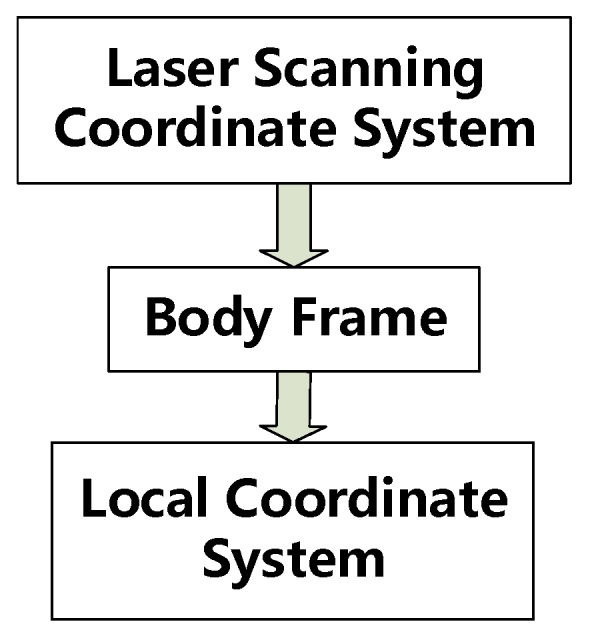
Procedure of the improved coordinate transformation.

**Figure 5 sensors-20-00645-f005:**
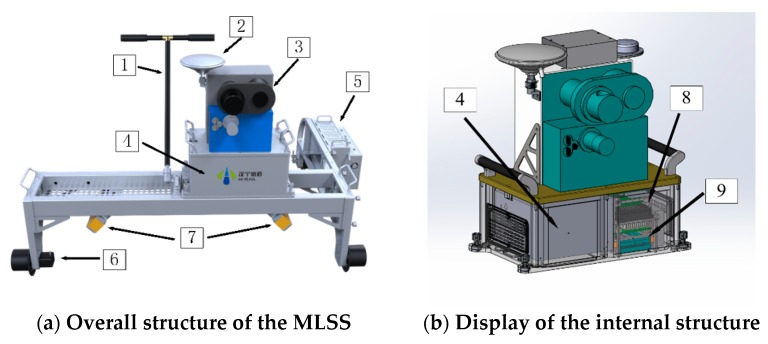
The structure information of the self-developed mobile laser scanning system (MLSS). (**a**) The overall structure of the self-developed mobile laser scanning system (MLSS). (**b**) Display of the internal structure. The names of the components in the instrument are as follows: 1. push rod; 2. GPS receiver antenna; 3. laser scanner; 4. inertial measurement unit (IMU) with three-axis gyroscopes and three-axis accelerometers (installed inside); 5. battery; 6. odometer; 7. 2-D laser scanner; 8. time synchronization control board; 9. industrial personal computer.

**Figure 6 sensors-20-00645-f006:**
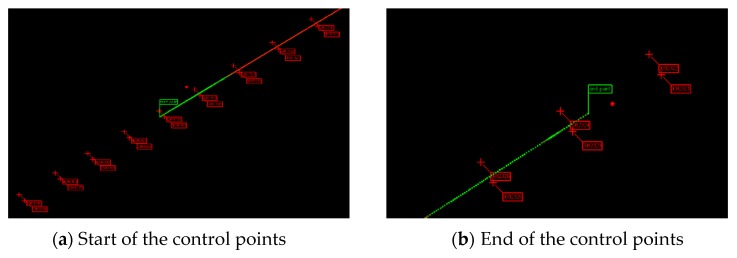
Trajectory and control points.

**Figure 7 sensors-20-00645-f007:**
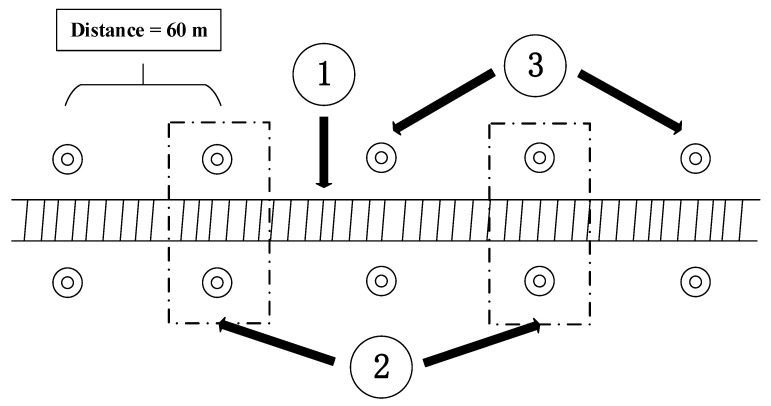
The strategy of using control points for correction. The meanings of each label in Figure 7 are as follows: 1. the railway or subway track; 2. control points used for correction; 3. control points used for verification.

**Figure 8 sensors-20-00645-f008:**
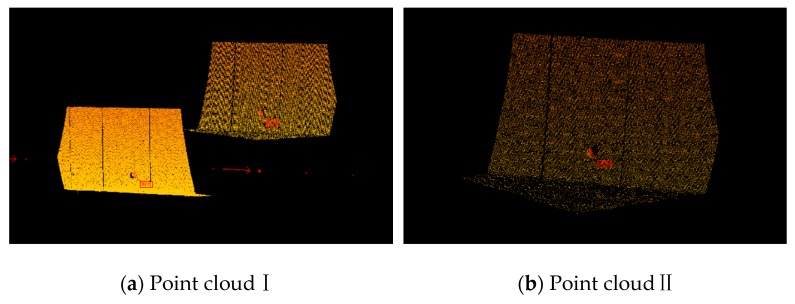
Extraction of control points from point clouds.

**Figure 9 sensors-20-00645-f009:**
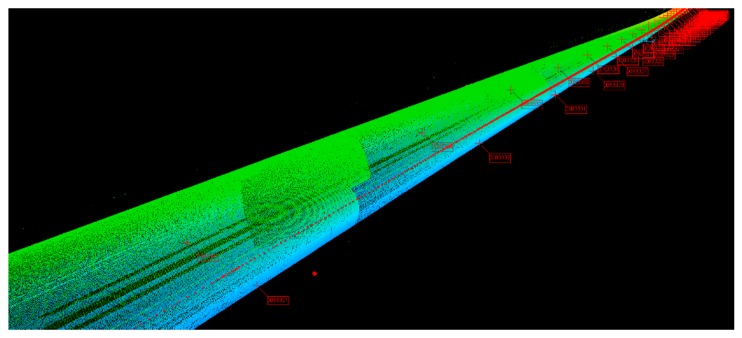
Point clouds of the railway tunnel and control points.

**Figure 10 sensors-20-00645-f010:**
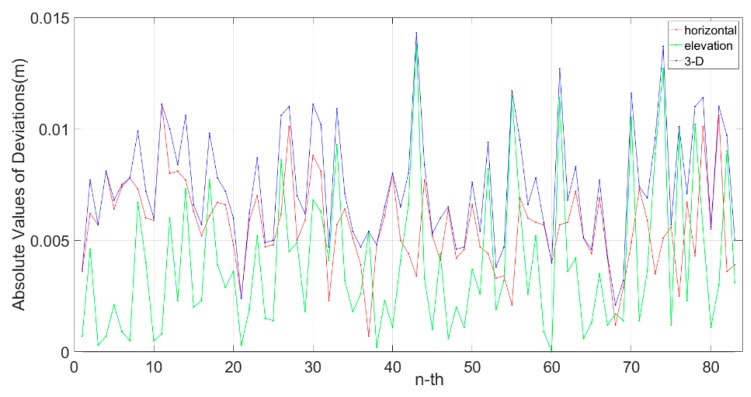
Absolute values of deviations between the coordinates of the control points.

**Figure 11 sensors-20-00645-f011:**
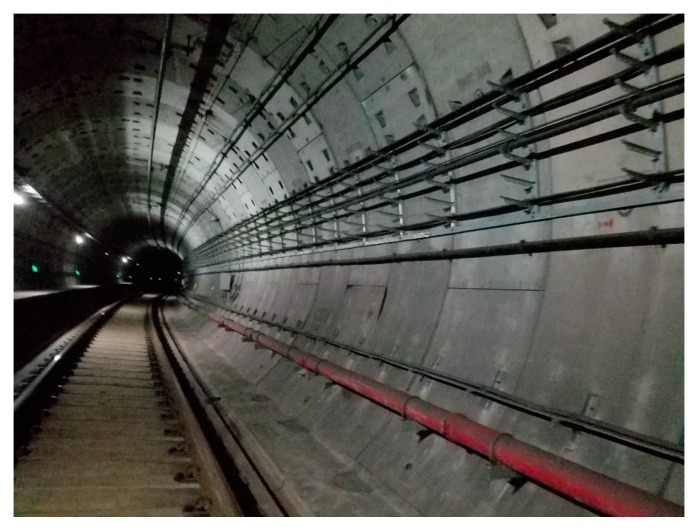
Practical working environment of the subway tunnel.

**Figure 12 sensors-20-00645-f012:**
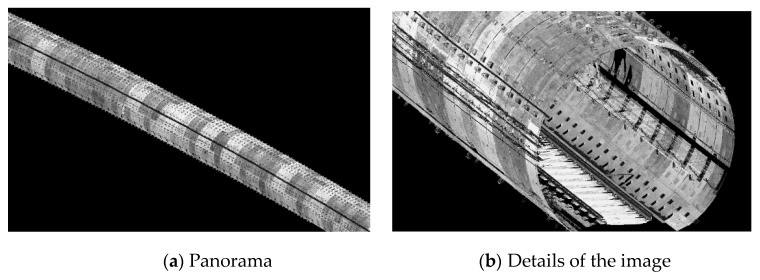
Point cloud image of the subway tunnel.

**Figure 13 sensors-20-00645-f013:**
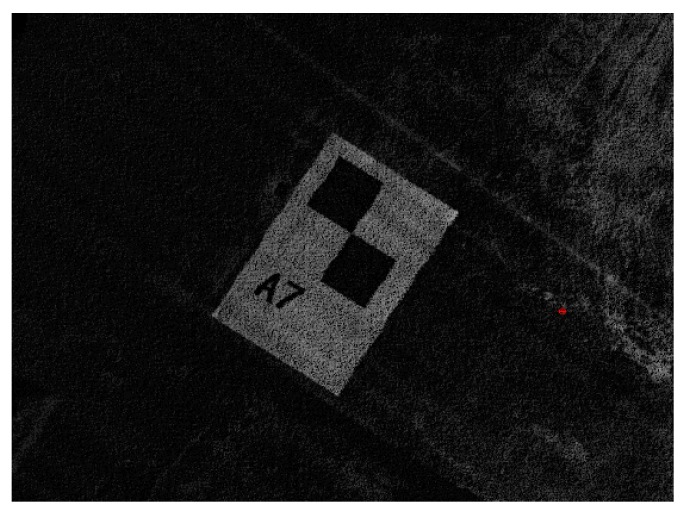
Point clouds of paper target.

**Table 1 sensors-20-00645-t001:** Performance of the laser scanner of the MLSS.

Emission Frequency	Scanning Frequency	Scanning Range	Measuring Distance	Distance Error (Reflectivity = 90%)	Efficiency
$10^{8}$ p/s	200 r/s	360°	0.5–119 m	2 mm (distance = 80 m)	3–5 km/h

**Table 2 sensors-20-00645-t002:** Performance of the IMU of the MLSS.

Gyro Bias	Gyro Bias Stability	Gyro Bias Repeatability	Gyro Random Walk	Accelerometer Bias	Accelerometer Bias Repeatability
≤±0.1°/h	≤0.01°/h	≤0.01°/h	≤0.003°/h1/2	≤0.00005 g	≤0.00005 g

**Table 3 sensors-20-00645-t003:** Absolute values of deviations between point clouds and the corresponding control points. RMS: root mean square error.

	Horizontal	Elevation	3-D
Maximum (m)	0.011	0.013	0.014
Minimum (m)	0.0007	0.000	0.002
RMS (m)	0.006	0.005	0.008

**Table 4 sensors-20-00645-t004:** Position accuracy of point clouds which are only corrected near the start and end points.

Horizontal Error (m)			Elevation Error (m)			3-D RMS (m)
Maximum	Average	RMS	Maximum	Average	RMS	
0.016	0.008	0.005	0.043	0.026	0.022	0.023

**Table 5 sensors-20-00645-t005:** Accuracy with control point correction in different densities.

Distance Interval (m)	Horizontal Error (m)			Elevation Error (m)			3-D RMS (m)
	Maximum	Average	RMS	Maximum	Average	RMS	
60	0.010	0.005	0.004	0.013	0.007	0.002	0.004
120	0.010	0.004	0.004	0.018	0.009	0.004	0.006
240	0.010	0.004	0.004	0.012	0.002	0.006	0.007
480	0.010	0.004	0.004	0.012	0.002	0.007	0.008

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
