# Peer review of "Absolute Positioning and Orientation of MLSS in a Subway Tunnel Based on Sparse Point-Assisted DR"

_sensors, 2020, doi:10.3390/s20030645_

Round 1
Reviewer 1 Report
Restricted expression of the title of this paper should be reorganized. Try to focus on your novel research. Please double-check that all abbreviations appearing for the first time have been described before or have not been described repeatedly. For instance, GEDO CE? What is four-parameter transformation method? Please describe it or provide proper reference. I have some puzzles about the pre-calibration between the GNSS and part of the inertial navigation parameters in an area of good GNSS signal. Why emphasize that it must be carried out in a straight track area? How long distance is the straight track segment required for calibration? Is it appropriate if there is a slight curvature change? Can the pre-verified parameters avoid the cumulative error divergence of the later inertial navigation system? Due to the use of GNSS for prior verification, is the statement about GNSS-denied appropriate in this paper? Is the inertial navigation system used in Table 2 based on a single-axis gyroscope or a three-axis gyroscope? Please add an analysis assessment in the experimental part to clarify the theoretical advantages described in parts of 2.5.1 and 2.5.2. What is the difference between the two examples for the test of the proposed theoretical method? Is it just a scene change or a distributed verification or something else?

Reviewer 2 Report
This manuscript is well written, and the results are very convincing.
Minor comments:
- Figure 4 should be referred to in the main body of the text, as done for all other figures.
- Numbers of Equations 8 and 9 ("(8), (9)", p.7) should be aligned with others.
- It is not clear, where the caption or description of Figure 8 ends, and main text continues (p.15).
- On p.11, 1st paragraph, a source is cited with the reference number 54 (“[53-54]”), which is not listed under the heading References. Suggesting this should be verified.
Author Response
Response to Reviewer 2 Comments
We appreciate very much the reviewer’s comments and good suggestions. We have revised this paper according to the decisions of the reviewers. The responses are as follows:
Reviewers 2:
Comments and Suggestions for Authors
This manuscript is well written, and the results are very convincing.
Minor comments:
This paper has been revised by a professional language organization, and modifications and more explanations has been made to the relevant areas.
Figure 4 should be referred to in the main body of the text, as done for all other figures.
Revised
Numbers of Equations 8 and 9 ("(8), (9)", p.7) should be aligned with others.
Revised
It is not clear, where the caption or description of Figure 8 ends, and main text continues (p.15).
Revised
The caption should be “Figure 8. Extraction of control points from point clouds.”
On p.11, 1st paragraph, a source is cited with the reference number 54 (“[53-54]”), which is not listed under the heading References. Suggesting this should be verified.
Revised
The quotation number of the last two citations are wrong and have been revised.
Round 2
Reviewer 1 Report
1. If DR technique is used to assist the sparse point? If yes, the title should be 'Absolute Positioning and Orientation of MLSS in a Subway Tunnel Based on Sparse Point Assisted with DR'.
2. Carefully check whether the quantitative indicators in the conclusion analysis lack the corresponding units, like, ‘The accuracy of the 3-D coordinates of the point clouds obtained can reach up to 4 mm when the distance interval of the control point for correction is 60 , and…’
3. Given the logic of the revised paper, the research work in this paper becomes clearer. However, in the field of engineering survey, the measurement accuracy of the level or total station is 0.1mm. That’s to say, only when the measurement accuracy reaches 0.1mm can the geometric deformation of the mm size of the structure be effectively feedbacked. Therefore, the present accuracy in this paper can only reach the level of mm, which is merely equal to the order magnitude of convergence deformation that is deserved to concern. So, how to define its applicable guidance in actual engineering? This problem is critical and should be clarified. The guidance value and positioning of this study for engineering applications need to be supplemented.
Author Response
Please see the attachment.

This manuscript is a resubmission of an earlier submission. The following is a list of the peer review reports and author responses from that submission.
Round 1
Reviewer 1 Report
This paper described a MLSS positioning and orientation method in subway detection by INS/Odometer. However:
Some of the English language is not writing in good style. such as, CPIII ? Lots of the symbol is not in one line as the sentences, earth should be "Earth", and so on; Fig.5 should be clearly point out the detailed experimental equipments, such as INS, MLSS and so on; The experimental parts only described both experiments, while no any analysis part about each experiments, which make the paper just a simple experimental report. How does the authors make the conclusion of the positioning precision of mm level ? Where is the reference system for the positioning ? The Table 4 gives the results of the system, and its precision to mm level, so the numeral results should be at least 4 bits after the dot. According to above consideration, this paper is on the border of decline and major revision, please revise this paper really carefully and improve the overall manuscirpt as much as possible.Author Response
Responses to Reviewer 1 Comments
We appreciate very much the reviewer’s comments and good suggestions. We have revised this paper according to the decisions of the reviewers. The responses are as follows:
Point 1: This paper described a MLSS positioning and orientation method in subway detection by INS/Odometer. However: Some of the English language is not writing in good style. Such as, CPIII ?
Response 1: This paper has been revised by a professional language organization. CPIII is the abbreviation of third-class control points in rail transportation, and we have added the full name where it first appeared.
Point 2: Lots of the symbol is not in one line as the sentences, earth should be "Earth", and so on;
Response 2: Checked the whole article and modified the formatting errors.
Point 3: Fig.5 should be clearly point out the detailed experimental equipments, such as INS, MLSS and so on;
Response 3: Some explanations are added to the components of the instrument.
Point 4 The experimental parts only described both experiments, while no any analysis part about each experiments, which make the paper just a simple experimental report.
Response 4: The structure of the experimental part has been adjusted, the language description has been modified, new explanations have been added to the pictures, and the analysis is added to make it easier to understand the results.
Point 5: How does the authors make the conclusion of the positioning precision of mm level?
Response 5: This paper utilizes part of the control point to correct the positions and attitudes of MLSS, the other part were left for the validation. The coordinates of point clouds is obtained through data fusion by using the positions and attitudes of MLSS, so the accuracy of point clouds can reflect the accuracy of positioning. All the control points is canned by MLSS, so their corresponding laser points can be found in the point cloud, the local coordinates of the control points (without been used before) were compared with the coordinates of their corresponding laser points.
Point 6: Where is the reference system for the positioning?
Response 6: Some explanations about the coordinate system were added in this paper. The solution of INS is in the navigation coordinate system (NCS, which is related to the position and orientation on the earth), and the coordinates of laser point is relative to the laser scanner coordinate system. But the coordinates of point clouds needs to be transfomed into local coordinate system. So there will be a lot of transformation operations and many steps of point cloud fusion, which is why we simplified the procedure in 2.5.
Point 7: The Table 4 gives the results of the system, and its precision to mm level, so the numeral results should be at least 4 bits after the dot.
Response 7: Because most of the coordinates of control points given to us 3 bits after the dot, we utilized the rounding algorithm and rounding a number up and down on existing decimals.
Point 8: According to above consideration, this paper is on the border of decline and major revision, please revise this paper really carefully and improve the overall manuscirpt as much as possible.
Response 8: Some changes have been made to the structure of the paper, added more explanations and modified the part of the algorithm.Some mistakes and formating errors were also modified. This paper has been revised by a professional language organization.

Reviewer 2 Report
The manuscript has to be better organized and clearly stated. Some parts may cause confusion, especially with so many equations. The method part and the equations have to be explained more clearly.
Please pay attention to the format of the manuscript (e.g. equation edit) and the statement in English. Make sure that each symbol in the equation is clearly explained and don't miss any symbol that may cause confusion. Please clear present each equation, for example, what are the measurements used, what is each element in the state vector, etc. More necessary statement is needed, eg. the definition of each coordinate system, what is body frame (IMU?), calibration method for the lever arm, how to do the synchronization of different sensors, etc. Section 2 has to be more clear about mechanization and error modeling. Since high-end IMU is applied, it is better to show the comparison of stand-alone IMU and IMU/odometer results. The figures need to be more clearly plotted and explained. For example, in Figure 9, what are the red and green represent and words are not clear at all. Some results are confusing, for example, Table 4 is accuracy without control points, is the result achieved by comparison of coordinated of control points in the point cloud and the true coordinates? Below the table, it is stated accuracy with the control point interval is 60, which causes confusion.Author Response
Responses to Reviewer 2 Comments
We appreciate very much the reviewer’s comments and good suggestions. We have revised this paper according to the decisions of the reviewers. The responses are as follows:
Point 1: The manuscript has to be better organized and clearly stated. Some parts may cause confusion, especially with so many equations. The method part and the equations have to be explained more clearly. Please pay attention to the format of the manuscript (e.g. equation edit) and the statement in English. Make sure that each symbol in the equation is clearly explained and don't miss any symbol that may cause confusion. Please clear present each equation, for example, what are the measurements used, what is each element in the state vector, etc. More necessary statement is needed, eg. the definition of each coordinate system, what is body frame (IMU?),
Response 1: This paper has been revised by a professional language organization. Some mistakes and formating errors were also modified. The structure of the experimental part has been adjusted, added more explanations and modified the part of the algorithm. New explanations have been added to the pictures and the symbols, and the analysis is added to make it easier to understand the results. In this paper, the body frame is the coordinate system of IMU, and we have added the explanations where it first appeared.
Point 2: Calibration method for the lever arm.
Response 2: Actuall, the calibration work should be completed when the instrument was built, it is not part of the algorithm in this paper, the method is simple and the related research is mature, but we have added the explanations : “In this paper, the calibration of ,, is carried out on an straight track with good GNSS signal, by comparing the positions and velocities calculated by INS/Odometer and GNSS,the exact values of ,, can be obtained. is obtained through the design parameters of the instrument.” In this paper.
Point 3: How to do the synchronization of different sensors, etc.
Response 3: The sensor space pose synchronization is carried out by calibration.
Time synchronizatio: The reference time of each sensor is obtained by the frequency division of 10MHz pulse generated by Complex Programmable Logic Device (CPLD).
The synchronization of different sensors should be completed when the instrument was built, it is not part of the algorithm in this paper, and it belongs to the field of hardware.
Point 4: Section 2 has to be clearer about mechanization and error modeling. Since high-end IMU is applied, it is better to show the comparison of stand-alone IMU and IMU/odometer results.
Response 4: Some modifications has been made to the principle part and some contents have been added, more explanations have been added to the equations and the symbols.
Subway detection work belongs to field of precision engineering measurement, and it requires absolute coordinates, even the high-end IMU cannot avoid error accumulation and becomes divergence, stand-alone IMU cannot even finish the detection work, its accuracy is far from IMU/odometer.
Point 5: The figures need to be more clearly plotted and explained. For example, in Figure 9, what are the red and green represent and words are not clear at all.
Response 5: Replaced some pictures, more explanations has been added to the pictures, equations and symbols. The analysis and discussions about the experimental results is also added to this paper.
Point 6: Some results are confusing, for example, Table 4 is accuracy without control points, is the result achieved by comparison of coordinated of control points in the point cloud and the true coordinates? Below the table, it is stated accuracy with the control point interval is 60, which causes confusion.
Response 6: Some misstatements has been corrected.
Table 4 shows the position accuracy of point clouds obtained by using the positions and attitudes of MLSS that without been corrected by the coordinates of control points. Table 5 shows the position accuracy of point clouds obtained by using MLSS’s positions and attitudes that been corrected by the coordinates of control points in different distance intervals.
This paper utilizes part of the control point to correct the positions and attitudes of MLSS, the other part were left for the validation. The coordinates of point clouds is obtained through data fusion by using the positions and attitudes of MLSS, so the accuracy of point clouds can reflect the accuracy of positioning. All the control points were canned by MLSS, so their corresponding laser points can be found in the point cloud, the local coordinates of the control points (without been used before) were compared with the coordinates of their corresponding laser points.
Some explanations about the coordinate system were added in this paper. The solution of INS is in the navigation coordinate system (NCS, which is related to the position and orientation on the earth), and the coordinates of laser point is relative to the laser scanner coordinate system. But the coordinates of point clouds needs to be transfomed into local coordinate system. So there will be a lot of transformation operations and many steps of point cloud fusion, which is why we simplified the procedure in 2.5.

Reviewer 3 Report
Attached the comments.

Author Response
Responses to Reviewer 3 Comments
We appreciate very much the reviewer’s comments and good suggestions. We have revised this paper according to the decisions of the reviewers. The responses are as follows:
Point 1: Using control points to improve the measurement accuracy is a very common way in surveying field, especially during the post processing (after measurement). With enough number of control points, by rotation, scale and shift, any measurement data can be corrected to control points. In addition, control point measurement is very time consuming. If the authors can make the measurement correction to the control points in real-time, the paper will be valuable. Apparently, in this paper, the accuracy improvement by control points was done during the post processing. Thus, the value of contribution is small.
Response 1:
First, the point cloud data fusion process cannot be finished in real-time. The laser scanner will produce a huge amount of data when building the spatial model of the object. The laser scanner used in this paper can collect one million points per second, each point has 3-D position information and attribute information, and it is just a very common laser scanner. The point cloud data fusion process cannot be finished in real-time. Additionally, for subway detection, real-time monitoring is meaningful less, the accuracy is more important, how to identify the problem of the subway with high precision is more important.
Second, even if we only use the point cloud around the control point to carry out the correction, the subsequent disease identification based on massive point cloud is also impossible to achieve in real time.
So why do we choose laser scanner as the sensor to build the spatial model of the object instead of other sensors like camera? Because the laser scanner has a very high measurement accuracy, which is unmatched by other sensors. The accuracy of the coordinate of laser point is 2mm when the measuring distance is 80m (under 90% reflectivity, and will be much better in subway detection work for the measuring distance is only several meters. The laser scanner is the most suitable instrument for subway detection. Traditional subway detection methods can not even complete the detection of some diseases, let alone the detection efficiency and reliability.
The algorithm in this paper is intended to make the MLSS get rid of the dependence on GNSS signals, and the positioning and orientation information can be used for point cloud data fusion to directly obtain the coordinates in LCS, this will greatly reduce the time for data processing and also improve the accuracy of point clouds.
The solution of INS is in the navigation coordinate system (NCS, which is related to the position and orientation on the earth), and the coordinates of laser point is relative to the laser scanner coordinate system. But the coordinates of point clouds needs to be transfomed into local coordinate system. So there will be a lot of transformation operations and many steps of point cloud fusion, which is why we simplified the procedure in 2.5.
This paper utilizes part of the control point to correct the positions and attitudes of MLSS, the other part were left for the validation. The coordinates of point clouds is obtained through data fusion by using the positions and attitudes of MLSS, so the accuracy of point clouds can reflect the accuracy of positioning. All the control points were canned by MLSS, so their corresponding laser points can be found in the point cloud, the local coordinates of the control points (without been used before) were compared with the coordinates of their corresponding laser points.
Point 2: In addition, the paper didn’t mention that how you placed the control points in an equal distance?
Response 2: The control points was not built by us, they were built at equal intervals by the railway construction department, and they belongs to railway infrastructure.
Point 3: How do you measure the control point coordinates?
Response 3: The coordinates of control points is provided by the railway construction department, while the MLSS passes by the control points distributed on both sides of the track, the local coordinates of control points were transmitted to the center of MLSS by using the ranging information of laser scanner.
Point 4: How do you detect the control points from point cloud (manually)?
Response 4: The paper target is placed next to the control point in order to make it convenient to find out the control point in the post-processing process.In order to save the time, only a small number of point clouds around the control points were fused according to the mileage information, and the laser points of corresponding control points can be easily find out with the asistance of paper target.
Point 5: Language needs to be improved; Some figures have no text description, e.g. Figure 8, 12
Response 5: This paper has been revised by a professional language organization. Some mistakes and formating errors were also modified. The structure of the experimental part has been adjusted, added more explanations and modified the part of the algorithm. New explanations has been added to the pictures and the symbols, and the analysis is added to make it easier to understand the results. In this paper, the body frame is the coordinate system of IMU, and we have added the explanations where it first appeared.
Point 6: Qingquan Li’ Who is it?
Response 6: Corrected the description.

Round 2
Reviewer 1 Report
This version of the manuscript is improved a lot when it compared with the previous manuscript.
Some of the parts should be described more clearly.
(1) The exprimental platform in Fig.5 seems is self-developed by which company ? While it looks like a matured platform from the outlook of the equipment. More inside figures about this platform ?
(2) The authors are still not clearly described the precision of the system is derived/computed ? The control points from one side can be used as the updating point, and on the other side can be used as the reference point. How many control points are utilized in the each expriments ? and How is the control points distributed along the scanned tunnel ? These questions should be clearly presented because for example in 200 meters' long tunnel, 10 control points or 50 control points for updating could obtain different results completely.
Therefore, the current format/content of the manuscript is still should be improved before publication.
Author Response
Please see the attachment。

Reviewer 2 Report
The paper is improved after revision. However, there are still typos and grammar issues. Some key figures are still not presented and explained (e.g. Fig 6). Please check the explaination to avoid confusion.
In the method section, the paper presents the integration of INS and Odometer. It seems that the point cloud coordinates are determined based on the attitude and position achieved by INS/odometer and the laser scanner is not applied in navigation. The result shows that the positioning accuracy of such a system can achieve about 2cm traveling 2km, which is not convincible. error sources and mitigation should be better explained.
Since the advantage of such a system proposed is sensor integration, comparison with individual sensor outcome is better to be included. Moreover, according to the result, the control point is applied to correct the estimation. Therefore, the corresponding method is better to be added in Section 2.
Author Response
Response to Reviewer 2 Comments
Point 1: The paper is improved after revision. However, there are still typos and grammar issues. Some key figures are still not presented and explained (e.g. Fig 6). Please check the explanation to avoid confusion.
Response 1: This paper has been revised by a professional language organization at its first major revision, and it has been checked by a native English speaking friend in this time. More explanations have been added to the relevant areas.
Point 2: In the method section, the paper presents the integration of INS and Odometer. It seems that the point cloud coordinates are determined based on the attitude and position achieved by INS/odometer and the laser scanner is not applied in navigation. The result shows that the positioning accuracy of such a system can achieve about 2cm traveling 2km, which is not convincible. Error sources and mitigation should be better explained.
Response 2: This is a descriptive mistake, which has been corrected in the article, which is shown in the following:
“Table 4 shows the position accuracy of the point clouds obtained by using the positions and attitudes of the MLSS that had been corrected only by the coordinates of the control points near the start and end points (we need two control points that are not paired to correct the estimation and provide absolute reference information), and there will be no correction elsewhere. The absolute accuracy of the 3-D coordinates was 2.3 cm, which shows the accuracy of the point clouds that can be obtained by using INS/odometer integrated navigation with no control point corrections in the middle of 1~2km. Obviously, the accuracy of the point clouds was not high enough.”
Point 3: Since the advantage of such a system proposed is sensor integration, comparison with individual sensor outcome is better to be included. Moreover, according to the result, the control point is applied to correct the estimation. Therefore, the corresponding method is better to be added in Section 2.
Response 3: We have added some explanations in Section 2, but the solution model of pure INS is too general and too mature, Section 2 described the Model of static initial alignment,the model of integrated navigation, the model of forward/reverse DR and the method of using control points to estimate the errors and transform the coordinates.
The method of using control points to estimate the errors and transform the coordinates is described in “2.5.3. Transformation from NCS to LCS”. Using the four parameters transformation method is like getting the trajectory from DR and the trajectory constrained by control points close to each other.
There is no GNSS signals in subway environment, and we need two control points that are not paired to correct the estimation and provide absolute reference information. Even if the control points used for correction are with the highest density, it is still impossible to use the INS only, so Table 4 and Table 5 shows the accuracy obtained with minimum number of control points used and the accuracy obtained with the correction of control points in different densities.

Reviewer 3 Report
Dear Authors,
After revision, the paper was well presented. Although there is room for further improvement of the method, this paper can be an initiate step toward the final goal of automation. As you mentioned that control points were provided by the third party, you couldn't select the types of control points. In the future, you may try to use a white mark as a control point. Thus, by an intensity value filtering, it is easy to recognize control points from millions of points within a second. It is very time consuming in post processing the control points manually. Automatic processing is needed.
Minor spell checking is needed. For example, in page 3, under figure1, 'an MLSS' should be 'a MLSS'. Please check the revision part carefully.
Author Response
Response to Reviewer 3 Comments
Point 1: After revision, the paper was well presented. Although there is room for further improvement of the method, this paper can be an initiate step toward the final goal of automation. As you mentioned that control points were provided by the third party, you couldn't select the types of control points. In the future, you may try to use a white mark as a control point. Thus, by an intensity value filtering, it is easy to recognize control points from millions of points within a second. It is very time consuming in post processing the control points manually. Automatic processing is needed.
Response 1: Thanks for the advice, we have added the following part in this paper:
“Besides, the filtering and feature recognition algorithm of point cloud can be further studied to assist the control point target recognition and disease extraction in the process of data post-processing, this can improve the work efficiency.”
Point 2: Minor spell checking is needed. For example, in page 3, under figure1, 'an MLSS' should be 'a MLSS'. Please check the revision part carefully.
Response 2: This paper has been revised by a professional language organization at its first major revision, and it has been checked by a native English speaking friend in this time. More explanations have been added to the relevant areas.
